# Geographical variations in maternal dietary patterns during pregnancy associated with birth weight in Shaanxi province, Northwestern China

Yini Liu[1], Huihui Zhang[1], Yaling Zhao[1], Fangyao Chen[1], Baibing Mi[1], Jing Zhou[2], Yulong Chen[3], Duolao Wang[4], Leilei Pei [ID][1] *

1 Department of Epidemiology and Health Statistics, School of Public Health, Xi'an Jiaotong University Health Science Center, Xi'an, Shaanxi, P.R. China, 2 Department of Pediatrics, The Second Affiliated Hospital of Xi'an Jiaotong University, Xi'an, Shaanxi, P.R. China, 3 Institute of Basic and Translational Medicine, Shaanxi Key Laboratory of Ischemic Cardiovascular Disease, Shaanxi Key Laboratory of Brain Disorders, Xi'an Medical University, Xi'an, Shaanxi, P.R. China, 4 Biostatistics Unit, Department of Clinical Sciences, Liverpool School of Tropical Medicine, Liverpool, United Kingdom

* pll_paper@126.com

**Data Availability Statement:** All relevant data are within the manuscript and its Supporting information files.

## Abstract

The geographical variation of maternal dietary patterns related to birth outcomes is important for improving the health of mothers and children; however, it is currently unknown. Thus, the objective of the study was to investigate geographical variations of maternal dietary pattern during pregnancy, and evaluate the spatial varying association of maternal dietary patterns in pregnancy with abnormal birth weight. A population-based cross-sectional study was conducted in Shaanxi province in Northwest China in 2013 to evaluate the relationship between abnormal birth weight and dietary pattern using the Geographically Weighted Logistic Regression (GWLR). Three dietary patterns during pregnancy were extracted through factor analysis, explaining approximately 45.8% of the variability of food intake. Approximately 81.6% of mothers with higher scores on the equilibrium pattern was more unlikely to have small for gestational age (SGA) infants, with the lower OR observed in Central and South Shaanxi. The snacks pattern was positively associated with low birth weight (LBW) for 23.2% of participants, with the highest OR in Central Shaanxi. Among about 80.0% of participants with higher scores on the snacks pattern living in South and Central Shaanxi, there was a higher risk for SGA. The OR values tend to descend from South to North Shaanxi. The OR values of the negative association between prudent pattern and LBW decreased from South to North Shaanxi among approximately 59.3% of participants. The prudent pattern was also negatively associated with the increasing risk of fetal macrosomia among 19.2% of participants living mainly in South Shaanxi. The association of maternal dietary patterns during pregnancy with abnormal birth weight varied geographically across Shaanxi province. The findings emphasize the importance of geographical distribution to improve the dietary patterns among disadvantaged pregnant women.

**Funding:** The study was supported by National Natural Science of Foundation of China (No. 81602928, 81541069) and Natural Science Foundation of Shaanxi Province (No. 2017JM8102). The funders had no role in study design, data collection and analysis, decision to publish, or preparation of the manuscript. We certify that all authors have not financial interest related to the work of this paper. The authors have declared that no competing interests exist. Meanwhile, there are no prior publications or submissions with any overlapping information, including studies and patients. We have reviewed the final version of the manuscript and approve it for publication. To the best of our knowledge and belief, this manuscript has not been published in whole or in part nor is it being considered for publication elsewhere.

**Competing interests:** The authors have declared that no competing interests exist.

## Introduction

Dietary patterns are useful to examine the quality of a person's overall diet related to health outcomes because it is acknowledged that consumption of individual food or nutrients does not occur in isolation but interacts with each other. Pregnancy is a critical period during which healthy diet characterized by consuming a variety of nutritious foods is essential for promoting and maintaining maternal and child health [1–4]. Because of socio-cultural and geographical reasons, maternal dietary patterns during pregnancy are different and population-specific in each area [5, 6]. Current studies regarding to the association of maternal dietary patterns during pregnancy with adverse birth outcomes have not always been consistent across regions [7, 8], which at least in part might have resulted from geographical heterogeneity. Geographical heterogeneity could cause biases for association estimates and their standard errors if they are not properly accounted for. Thus far, however, there was few evidence demonstrating the geographical difference in the relationship between maternal dietary patterns during pregnancy and birth outcomes of infants. Therefore, it is necessary to find an efficient way to uncover such geographically varying characteristics of maternal dietary patterns during pregnancy that are associated with birth outcomes, which would facilitate the identification of links between diet and adverse birth outcomes.

Shaanxi located in Northwest China, is one of the provinces with huge variations in geographical environment and dietary structure in different areas. Thus, geographical heterogeneity cannot be ignored when we study the spatial variation of maternal dietary patterns during pregnancy in relation to birth outcomes. Abnormal birth weight (e.g. low BW or high BW) representing an important adverse birth outcome is strongly correlated with various health outcomes including neonatal mortality and morbidity as well as developing chronic diseases throughout their life [9–12]. In previous studies, conventional global regression models were commonly used to explore the association of maternal dietary pattern during pregnancy with abnormal birth weight. The traditional global regression which ignores the impacts of local variations produces average parameters to reflect the geographical relationships equally over the whole study area. Due to the lack of consideration in geographical heterogeneity, evaluation by conventional global regression is inadequate. To address this issue, geographically weighted regression (GWR) model could be applied to explore how one dependent variable changes in response to one or more independent variables at the local scale. Through taking the samples within a defined neighborhood into calculation by giving more weights to nearby samples than those further away, GWR generates local coefficients to elucidate the relationships within the neighborhood. Using maps to show the geographical variation in the relative importance of each covariate, GWR provides a visually powerful way of conveying the observations to the public health practitioners.

In Shaanxi province of Northwest China, to the best of knowledge, how to evaluate the geographical variation of maternal dietary patterns that are related to birth outcomes are not known but are important for improving the health of mothers and children. Thus, to address the aforementioned research questions, the objectives of this study were twofold: (1) to investigate geographical variations of maternal dietary pattern during pregnancy; (2) use GWR to evaluate the geographically varying association of maternal dietary patterns during pregnancy with abnormal birth weight.

## Materials and methods

### Study design and participants

From August to November 2013, a large-scale cross-sectional survey was conducted in Shaanxi province, in Northwest China with the purpose to assess the birth outcomes. The subjects of

the study were those live births born from January 2010 to November 2013 and their mothers. Participants were obtained by means of multi-stage random sampling method, that is, district random sampling was carried out after urban and rural areas were stratified to control for the impact of uneven distribution of dietary patterns on the risk of birth outcomes in different regions. In China, the administrative structure is composed of three layers, county, township and village in rural areas and district, street and community in urban areas, respectively. Based on the proportion of urban and rural areas, and considering the population density and fertility level, 20 counties in rural areas and 10 districts in urban areas were randomly selected by proportionate probability size (PPS) method. From each selected counties, 6 villages each from 6 townships were randomly selected. In urban areas, however, from each selected districts, 6 communities from each of 3 selected streets were sampled randomly. Finally, 10 babies born during 2010–2013 and their mothers in each chosen villages and 20 in each community were selected by a completely random sampling method. A total of 7,934 eligible women who completely fulfilled the diet survey were included into the study. The study was conducted in accordance with the Helsinki Declaration and was approved and reviewed by the Ethics Committee of Xi'an Jiaotong University Health Science Center (approval code: 2012008) [13].

## Data collection

A structured questionnaire was used to obtain all information through face-to-face interview of mothers of the neonates. Firstly, socio-demographic characteristics were obtained by a standard family questionnaire. Secondly, dietary intakes of mothers during the entire pregnancy was assessed by a semi-quantitative food frequency questionnaire (FFQ). It has been demonstrated that FFQ in the survey has satisfied reliability and validity for pregnant women in rural western China [14, 15]. In the FFQ, there are 102 food items, including fourteen food groups (cereals, potatoes, legumes, vegetables, fungi, fruits, nuts, meats, dairy, eggs, fish, ethnic snacks, fast foods, and beverages) based on the China Food Composition Table [16]. The frequency of 102 food items was divided into eight categories ranging from "never" to "$\geq$ 2 times/day". With the assistance of food portion images, the portion sizes for the 102 food items were recorded [14, 15]. The information of neonates including birth weight, birth date, and so on, was obtained from medical records.

## Quality control

Field surveys and data collection were performed by the trained field staff from Xi'an Jiaotong University Health Science Center. To control the quality of the data, all questionnaires and procedures were pre-tested in field prior to the formal survey, and then detailed interview guidelines were formulated. A total of 10 teams from Xi'an Jiaotong University Health Science Center performed the surveys in the selected counties or districts. Each team was composed of 10–12 qualified investigators, a supervisor from, and a pediatrician from the local maternal and child health hospital. During the field survey, the supervisors monitored all field work and reviewed the questionnaire. When errors or missing items were identified, participants were re-interviewed immediately. With sustained support from local hospitals, health administration departments and the Provincial Ministry of Health, we collected all information within the local village clinics and community health service centers [13].

## Study variables

Abnormal birth weight including low birth weight (LBW, birth weight < 2500 g), macrosomia (birth weight > 4000 g), small for gestational age (SGA, birth weight was below the 10th percentile of the average gestational age) and large for gestational age (LGA, birth weight was

above the 90th percentile of the average gestational age) was considered as final outcome variables in the multivariate analysis [17, 18]. As the independent variable, dietary pattern had two dummy variables: T1 (1 = yes, 0 = no) and T3 (1 = yes, 0 = no), with T2 as a reference group. The socio-demographic variables include child gender (male, female), fetal number (singleton, twin and multi-fetal), infant parity (1, 2, or ≥ 3), childbearing age (18–24, 25–29, ≥ 30 years), mother's education (primary school and below, junior high school, senior high school, college and above), mother's residence during pregnancy (floating, permanent), Household wealth Index (poor, middle-income, rich), altitude of residence ($< 500$, 500–1000, $\geq 1000$), and area (South Shaanxi, Central Shaanxi, North Shaanxi).

## Statistical analysis

Epidata 3.1 software with double entry was used to establish the database. Maternal dietary patterns (equilibrium, snack, and prudent) were derived by principal component analysis (PCA) and varimax rotation method using SPSS software. Before deriving the dietary patterns, the reliability of PCA was verified by Kaiser–Meyer–Olkin value and Bartlett's test of sphericity. The number of dietary patterns was calculated by eigenvalues ($>1.2$), variance contribution rate ($>5\%$), the scree plot and the interpretability of the factors [19, 20]. Food groups with absolute factor loadings $>0.3$ were considered to be highly correlated to the identified factors. Factor scores were calculated for each mother to indicate if mother's diet was consistent with the respective dietary patterns.

The vector map of China available at the National Platform for Common Geospatial Information Services (http://www.tianditu.com/) was used to obtain the geographic information (including geographic coordinates and altitude) for each township/street in Shaanxi province. The basic geographic unit in the study was townships and streets. Geographic coordinates (i.e., latitude/longitude) of the townships and streets was used to define their geographic location.

The GWLR model was employed to determine the relationship between abnormal birth weight and individual factors using the formula below [21]:

$$\log\left(\frac{P(y_i = 1)}{1 - P(y_i = 1)}\right) = \beta_{0i}(u_i, v_i) + \sum_{j=1}^{k}\left(\beta_{ij}(u_i, v_i)x_{ij}\right)$$

The equation hypothesizes that $y_i$ (dependent variables) was any abnormal birth weight for each individual i, $x_{ij}$ represented a set of independent variables($j = 1, \ldots, k$) for the individual i, ($u_i, v_i$) was regarded as the x-y coordinates of the individual i; $\beta_{ij}$ (regression coefficient) was deemed as the estimated effect of independent variable j for the individual i. The GWLR model is an extension and deformation of the typical linear regression model. The geographical data was incorporated into the model to create conditions for analyzing the geographical characteristics of the regression relationship.

We selected the GWR 4.0 software in this study (https://geodacenter.asu.edu/software/downloads/gwr_downloads) and the iterative reweighted least squares method was used to establish the GWLR model. During the process of modeling, a distance-based weighting scheme was used to assign weights to each township/street and the GWLR equation was estimated for each township/street. The kernel type and function for geographic weighting was used to estimate local coefficients for each township/street and bandwidth size was adaptive bisquare. Corrected Akaike Information Criterion (AICc) was used as the golden section search method to select the bandwidth of the adaptive kernel. The variability of each coefficient in geographic location was tested by comparing the difference [22]. In the GWLR model, OR corresponding to the unit change in the variables was obtained by exponentiation of the explanatory variable coefficients [22]. The variability of the influence of abnormal birth weight

on individuals in different regions was evaluated by comparing the variability of each coefficient in each geographic location [23].

The analysis was completed by the following steps. Firstly, the effects of each dietary pattern on abnormal birth weight were determined by non-spatial logistic regression analysis using IBM SPSS Statistics 25.0. For bilateral test, the test level was set at the level of α = 0.05. Subsequently, all explanatory variables were entered into the GWLR model as described above. The variables with the largest p-value were gradually eliminated from the model until only those with local p-value <0.1 were obtained. Finally, ArcGIS 10.0 software (Environmental Systems Research Institute, Inc., Redlands, CA, US) was used to show the local average estimates (OR) and p-values of individuals on the Shaanxi map. These values were used to intuitively reflect the geographical differences in the relationship between abnormal birth weight outcome and individual factors.

## Results

### Baseline characteristics of the participants

In this study, a total of 173 towns/streets were chosen within 30 counties in Shaanxi province, and 7,934 live infants were included in the final analysis. Among the mothers included in the study, the average age was 26.88±4.62 years and the average parity was 1.44 ± 0.56. The proportions of study participants from northern, central and southern Shaanxi were 18.1, 51.1 and 30.7%, respectively. From south to north, Shaanxi is divided into three physiographic regions by the North Mountains and the Qin Mountains. These three parts are North Shaanxi, Central Shaanxi, and South Shaanxi. North Shaanxi is a plateau, central Shaanxi is a plain and south Shaanxi is a mountainous region with an elevation of about 1500–3000 meters. Of the infants, the average age was 8.92 ±6.30 months (range 0–45 months), 53.6% were male and 98.8% were singletons.

### Characteristics of dietary pattern distribution during pregnancy among mothers

Through principal component analysis (PCA), three dietary patterns during pregnancy were extracted in Shaanxi province, northwest China, which explained approximately 45.8% of the variability in food intake (28.4%, 9.5% and 7.9% for three dietary patterns, respectively). According to the high positive loadings from various foods, including vegetables, homonemeae, beans meats, meat or flesh, fruit, fish and shrimp, potato, nuts, dairy, fast-food, sweetmeat, cereal, eggs and beverages, this first pattern was labelled as the 'equilibrium pattern'. The highest and lowest scores of the equilibrium pattern were in the Central Shaanxi and Northern Shaanxi, respectively. (Fig 1). The second pattern labelled as 'snacks pattern' had high loading from beverages, sweetmeat, fast-food, dairy and eggs. The scores of the snacks pattern were the higher in North Shaanxi, followed by South Shaanxi and Central Shaanxi (Fig 2). The third pattern with high intakes of dairy products and eggs was labelled as the 'prudent pattern' (Table 1). The scores of the prudent pattern had decreasing trend of from south to north. (Fig 3).

Table 2 showed the sociodemographic characteristics and median daily intakes of energy and nutrients according to tertiles of dietary patterns scores among pregnant women in Shaanxi province. In terms of equilibrium pattern, pregnant women with high scores were more likely to live in south or central Shaanxi, areas with lower altitudes (<500 meters) and be better educated and wealthier. Besides, mothers with high scores on equilibrium pattern tend to have higher childbearing age and lower parity. The sociodemographic characteristics of the

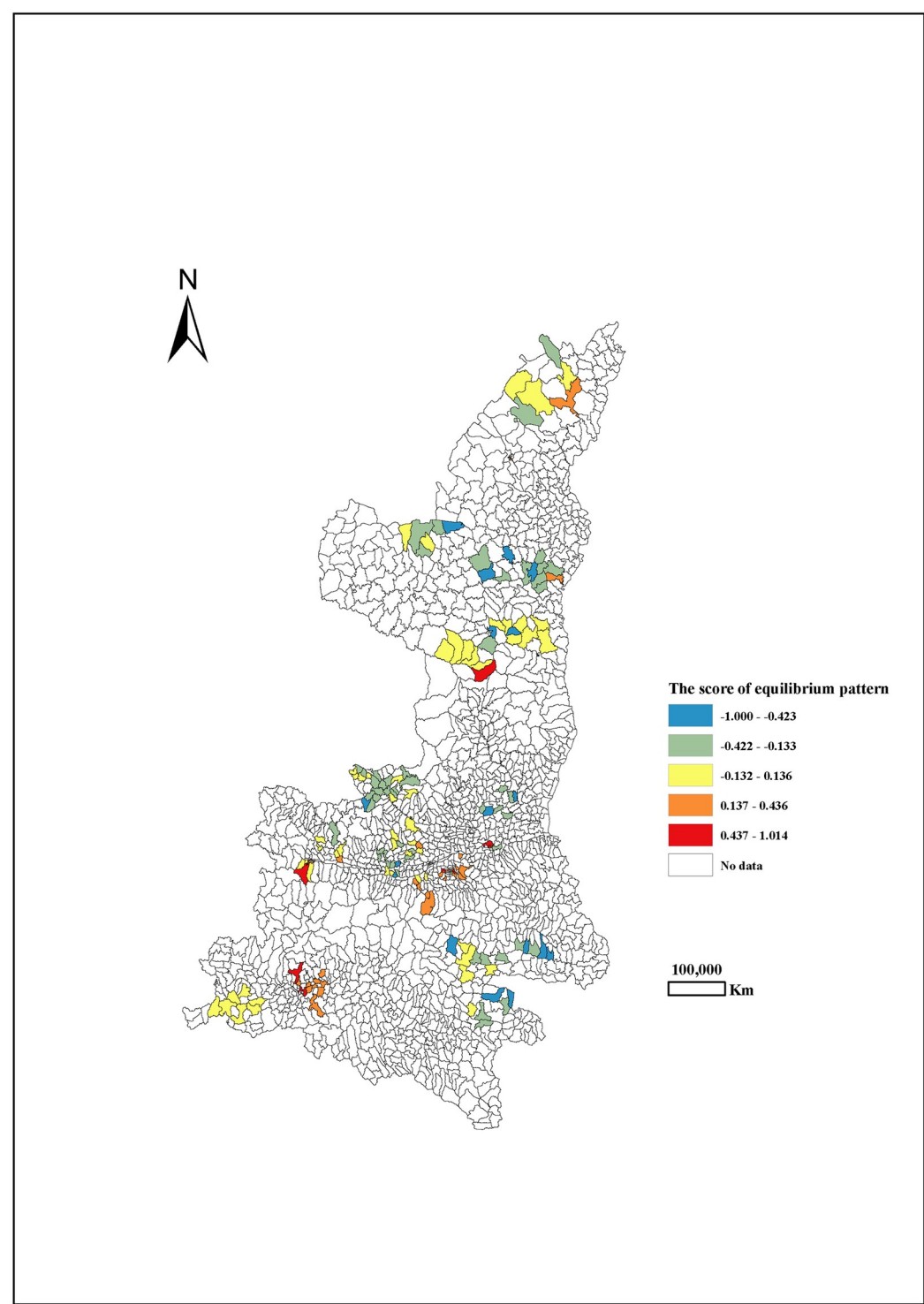

**Fig 1. The score of equilibrium pattern among pregnant women by different areas in Shaanxi province, 2013.**

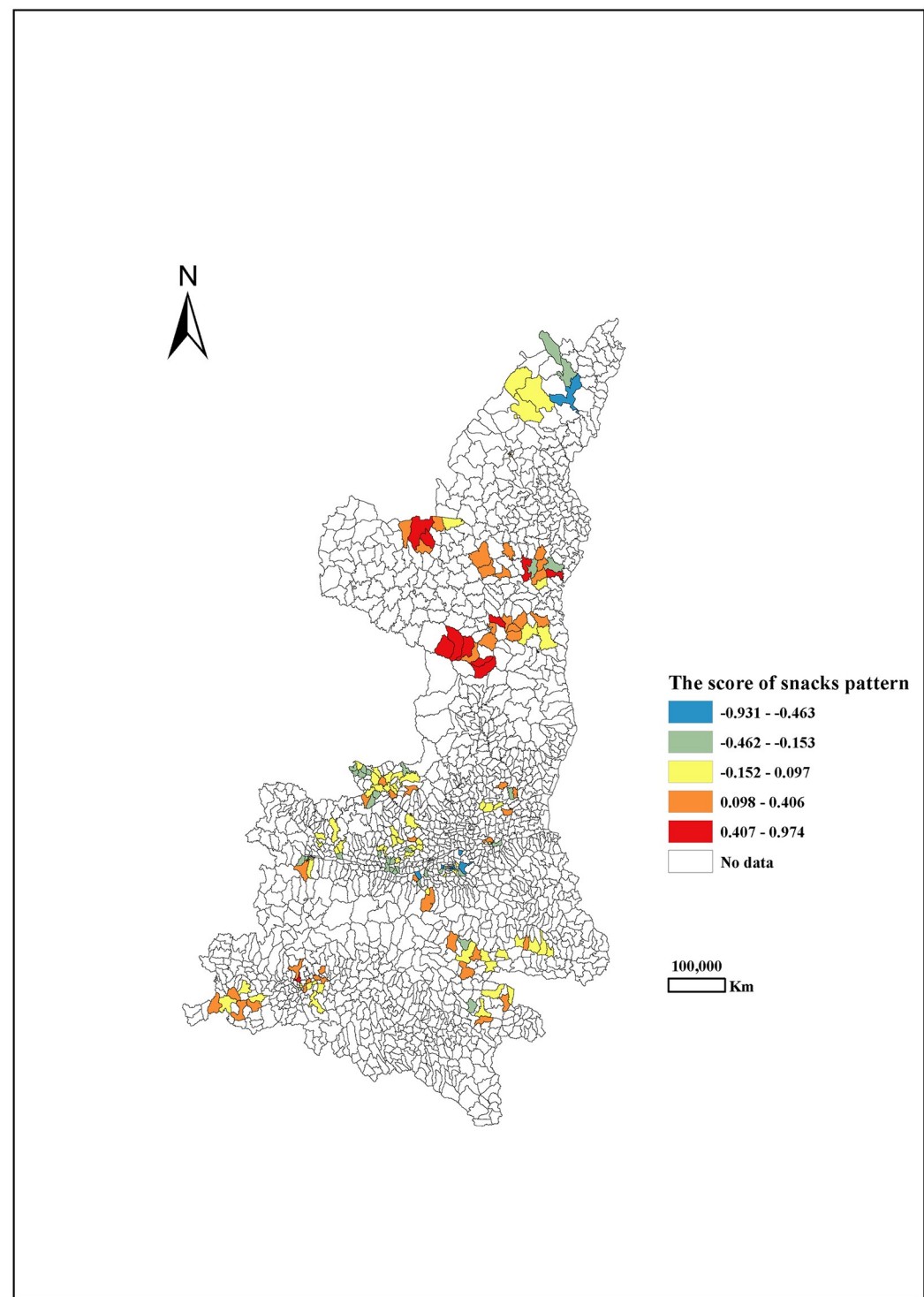

**Fig 2. The score of snacks pattern among pregnant women by different areas in Shaanxi province, 2013.**

**Table 1. Factor loadings for the identified dietary patterns among pregnant women (n 7934) in Shaanxi province, Northwest China, 2013.**

| Dietary pattern | Food group | Factor loading* | Cumulative variance explained (%) |
|---|---|---|---|
| Equilibrium pattern | vegetable | 0.762 | 28.4 |
| | homonemeae | 0.695 | |
| | beans | 0.624 | |
| | meat or flesh | 0.602 | |
| | fruit | 0.590 | |
| | fish and shrimp | 0.537 | |
| | potato | 0.535 | |
| | nuts | 0.504 | |
| | dairy | 0.469 | |
| | fast-food | 0.469 | |
| | sweetmeat | 0.443 | |
| | cereal | 0.357 | |
| | eggs | 0.338 | |
| | beverages | 0.315 | |
| Snacks pattern | beverages | 0.641 | 37.9 |
| | sweetmeat | 0.574 | |
| | fast-food | 0.348 | |
| | dairy | -0.333 | |
| | eggs | -0.437 | |
| Prudent pattern | eggs | 0.408 | 45.8 |
| | dairy | 0.331 | |
| | vegetable | -0.342 | |
| | potato | -0.534 | |

*Food groups are listed by the size of factor loadings, with negative loadings at the end. Food groups with absolute factor loadings <0.300 are not shown.

mothers with prudent pattern was similar to those with equilibrium pattern, except for maternal childbearing age which did not differ between tertiles of prudent pattern scores. On the contrary, mothers with high scores on the snacks pattern were more likely to live in north Shaanxi, areas with higher altitudes (>1000 meters), be less educated and poor, and be aged 18–24 years at delivery. Women with higher scores on the equilibrium pattern had higher intakes of energy and all selected nutrients, while those with lower scores on the prudent pattern and the snacks pattern had higher energy intake, carbohydrate intake and higher intakes of other selected nutrients.

## Characteristics of abnormal birth weight

Among the live infants, the prevalence of abnormal birth weight was 3.7% for LBW, 6.2% for macrosomia, 10.9% for SGA, 10.0% for LGA, respectively. The prevalence of LBW and LGA were the highest in South Shaanxi (4.1% for LBW and 10.3% for SGA, respectively), followed by Central Shaanxi (3.7% and 10.2%) and North Shaanxi (3.3% and 10.1%) (S1 and S2 Figs). The incidence of macrosomia was 6.6% in North Shaanxi, 6.2% in Central Shaanxi and 6.1% in Southern Shaanxi respectively (S3 Fig). SGA incidence in Central Shaanxi (11.7%) was higher than that in South Shaanxi (10.9%) or Northern Shaanxi (10.0%). (S4 Fig). The prevalence of abnormal birth weight differed across the baseline characteristics, and are presented in Table 3. The incidence of LBW was significantly lower in T3 group of equilibrium pattern, T1 group of snacks pattern, and T1 or T3 group of prudent pattern, respectively (P<0.05).

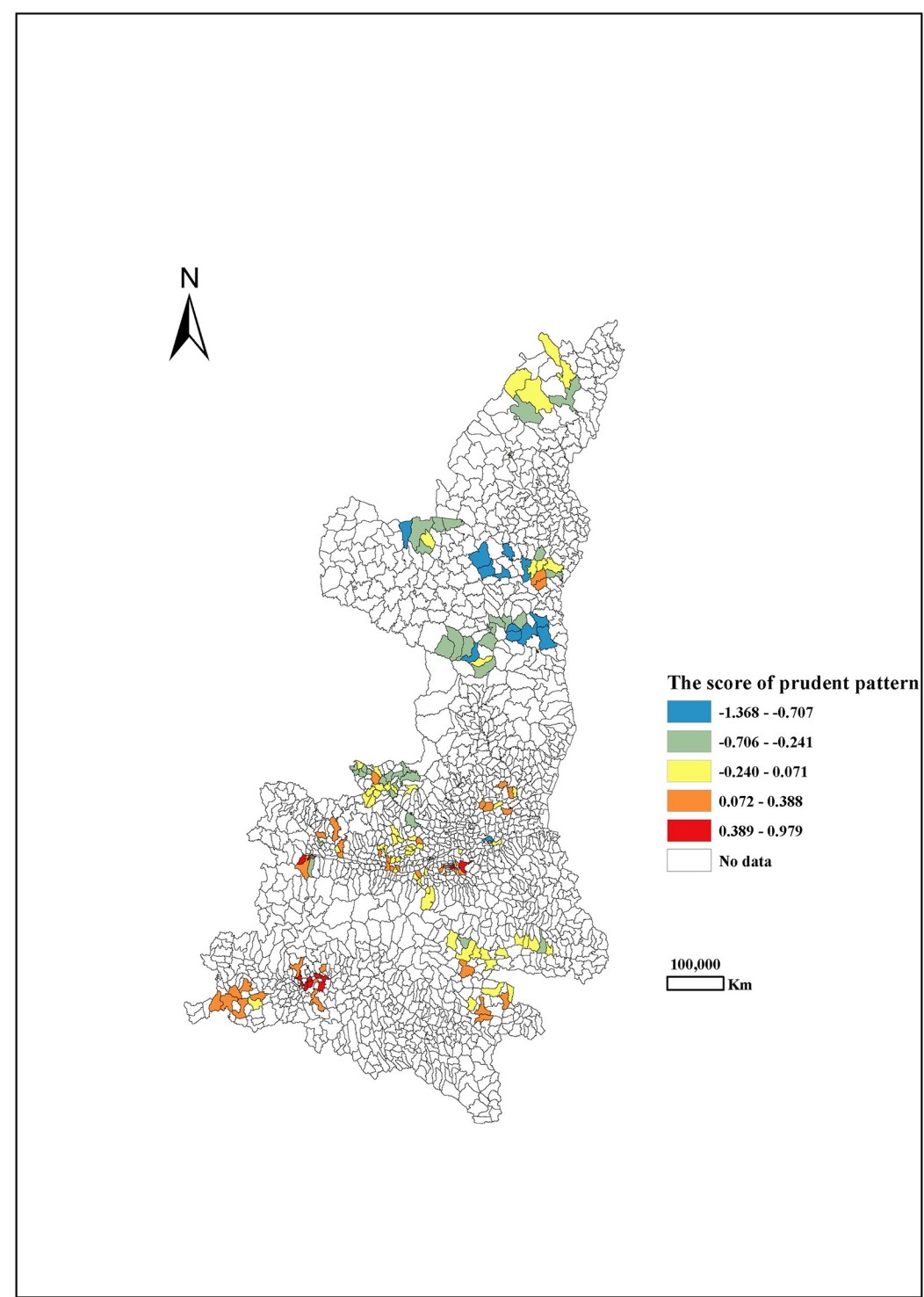

**Fig 3. The score of prudent pattern among pregnant women by different areas in Shaanxi province, 2013.**

**Table 2. The characteristics of dietary patterns among pregnant women (n 7934) in Shaanxi province, Northwest China, 2013.**

| | Equilibrium pattern | | | | Snacks pattern | | | | Prudent pattern | | | |
|---|---|---|---|---|---|---|---|---|---|---|---|---|
| | T1 | T2 | T3 | P* | T1 | T2 | T3 | P* | T1 | T2 | T3 | P* |
| *Sociodemographic characteristics*[†] | | | | | | | | | | | | |
| Child gender (%) | | | | 0.490 | | | | 0.953 | | | | 0.244 |
| Male | 1393 (32.7) | 1425 (33.5) | 1438 (33.8) | | 1423 (33.4) | 1418 (33.3) | 1415 (33.2) | | 1445 (34.0) | 1386 (32.6) | 1425 (33.5) | |
| Female | 1248 (34.0) | 1216 (33.1) | 1210 (32.9) | | 1218 (33.2) | 1224 (33.3) | 1232 (33.5) | | 1194 (32.5) | 1256 (34.2) | 1224 (33.3) | |
| Fetal number (%) | | | | 0.051 | | | | 0.746 | | | | 0.809 |
| Singleton | 2614 (33.4) | 2598 (33.2) | 2625 (33.5) | | 2610 (33.3) | 2607 (33.3) | 2620 (33.4) | | 2611 (33.3) | 2611 (33.3) | 2615 (33.4) | |
| Twin and multi-fetal | 28 (29.2) | 43 (44.8) | 25 (26.0) | | 32 (33.3) | 35 (36.5) | 29 (30.2) | | 30 (31.3) | 31 (32.3) | 35 (36.5) | |
| Infant parity (%) | | | | <0.001 | | | | <0.001 | | | | <0.001 |
| 1 | 1436 (30.8) | 1524 (32.7) | 1705 (36.5) | | 1710 (36.7) | 1456 (31.2) | 1499 (32.1) | | 1388 (29.8) | 1524 (32.7) | 1753 (37.6) | |
| 2 | 1100 (37.1) | 1010 (34.1) | 856 (28.9) | | 851 (28.7) | 1074 (36.2) | 1041 (35.1) | | 1115 (37.6) | 1030 (34.7) | 821 (27.7) | |
| ≥3 | 96 (37.1) | 94 (36.3) | 69 (26.6) | | 72 (27.8) | 98 (37.8) | 89 (34.4) | | 128 (49.4) | 70 (27.0) | 61 (23.6) | |
| Childbearing age (%) | | | | <0.001 | | | | <0.001 | | | | 0.201 |
| 18–24 | 988 (36.3) | 885 (32.5) | 848 (31.2) | | 776 (28.5) | 860 (31.6) | 1085 (39.9) | | 917 (33.7) | 924 (34.0) | 880 (32.3) | |
| 25–29 | 987 (30.4) | 1127 (34.7) | 1138 (35.0) | | 1176 (36.2) | 1090 (33.5) | 986 (30.3) | | 1051 (32.3) | 1100 (33.8) | 1101 (33.9) | |
| ≥30 | 649 (34.3) | 605 (32.0) | 639 (33.8) | | 668 (35.3) | 670 (35.4) | 555 (29.3) | | 656 (34.7) | 595 (31.4) | 642 (33.9) | |
| Mother's education (%) | | | | <0.001 | | | | <0.001 | | | | <0.001 |
| Primary school and below | 306 (42.1) | 257 (35.4) | 163 (22.5) | | 154 (21.2) | 273 (37.6) | 299 (41.2) | | 317 (43.7) | 250 (34.4) | 159 (21.9) | |
| Junior high school | 1589 (37.3) | 1422 (33.4) | 1251 (29.4) | | 1212 (28.4) | 1491 (35.0) | 1559 (36.6) | | 1525 (35.8) | 1466 (34.4) | 1271 (29.8) | |
| Senior high school | 515 (29.5) | 570 (32.6) | 661 (37.9) | | 654 (37.5) | 564 (32.3) | 528 (30.2) | | 503 (28.8) | 586 (33.6) | 657 (37.6) | |
| College and above | 225 (19.0) | 391 (33.1) | 566 (47.9) | | 614 (51.9) | 310 (26.2) | 258 (21.8) | | 293 (24.8) | 335 (28.3) | 554 (46.9) | |
| Mother's residence during pregnancy (%) | | | | 0.141 | | | | 0.001 | | | | <0.001 |
| Floating | 311 (30.7) | 340 (33.6) | 361 (35.7) | | 385 (38.0) | 331 (32.7) | 296 (29.2) | | 270 (26.7) | 328 (32.4) | 414 (40.9) | |
| Permanent | 2309 (33.6) | 2285 (33.3) | 2277 (33.1) | | 2246 (32.7) | 2292 (33.4) | 2333 (34.0) | | 2356 (34.3) | 2289 (33.3) | 2226 (32.4) | |
| Household wealth Index (%) | | | | <0.001 | | | | <0.001 | | | | <0.001 |
| Poor | 948 (36.9) | 923 (35.9) | 698 (27.2) | | 758 (29.5) | 911 (35.5) | 900 (35.0) | | 969 (37.7) | 885 (34.4) | 715 (27.8) | |
| Middle-income | 847 (32.2) | 871 (33.1) | 912 (34.7) | | 912 (34.7) | 876 (33.3) | 842 (32.0) | | 881 (33.5) | 862 (32.8) | 887 (33.7) | |
| Rich | 847 (31.0) | 848 (31.0) | 1040 (38.0) | | 972 (35.5) | 855 (31.3) | 908 (33.2) | | 792 (29.0) | 895 (32.7) | 1048 (38.3) | |
| Altitude (%) | | | | <0.001 | | | | <0.001 | | | | <0.001 |
| <500 | 667 (28.3) | 751 (31.9) | 937 (39.8) | | 1003 (42.6) | 715 (30.4) | 637 (27.0) | | 539 (22.9) | 764 (32.4) | 1052 (44.7) | |
| 500–1000 | 1265 (34.8) | 1161 (32.0) | 1207 (33.2) | | 1146 (31.5) | 1248 (34.4) | 1239 (34.1) | | 1125 (31.0) | 1226 (33.7) | 1282 (35.3) | |
| >1000 | 710 (36.5) | 730 (37.5) | 506 (26.0) | | 493 (25.3) | 679 (34.9) | 774 (39.8) | | 978 (50.3) | 652 (33.5) | 316 (16.2) | |
| Area (%) | | | | <0.001 | | | | <0.001 | | | | <0.001 |
| South area | 863 (35.4) | 716 (29.4) | 860 (35.3) | | 732 (30.0) | 783 (32.1) | 924 (37.9) | | 622 (25.5) | 801 (32.8) | 1016 (41.7) | |
| Central area | 1288 (31.8) | 1346 (33.2) | 1422 (35.1) | | 1623 (40.0) | 1383 (34.1) | 1050 (25.9) | | 1194 (29.4) | 1430 (35.3) | 1432 (35.3) | |
| North area | 491 (34.1) | 580 (40.3) | 368 (25.6) | | 287 (19.9) | 476 (33.1) | 676 (47.0) | | 826 (57.4) | 411 (28.6) | 202 (14.0) | |

*(Continued)*

**Table 2.** (Continued)

| | Equilibrium pattern | | | | Snacks pattern | | | | Prudent pattern | | | |
|---|---|---|---|---|---|---|---|---|---|---|---|---|
| | T1 | T2 | T3 | P* | T1 | T2 | T3 | P* | T1 | T2 | T3 | P* |
| *Median daily intake* | | | | | | | | | | | | |
| Energy(KJ) | 4417 | 6681 | 10120 | <0.001 | 8329 | 5933 | 6011 | <0.001 | 7375 | 5930 | 6925 | <0.001 |
| Protein(g) | 39 | 61 | 97 | <0.001 | 82 | 53 | 49 | <0.001 | 65 | 53 | 66 | <0.001 |
| Fat(g) | 20 | 38 | 69 | <0.001 | 54 | 31 | 30 | <0.001 | 35 | 32 | 48 | <0.001 |
| Carbohydrate(g) | 195 | 276 | 388 | <0.001 | 320 | 254 | 260 | <0.001 | 325 | 251 | 255 | <0.001 |
| Vitamin A(μg RE) | 169 | 392 | 794 | <0.001 | 500 | 325 | 387 | <0.001 | 523 | 306 | 380 | <0.001 |
| Thiamin(mg) | 0.4 | 0.6 | 1 | <0.001 | 0.8 | 0.6 | 0.6 | <0.001 | 0.8 | 0.6 | 0.6 | <0.001 |
| Riboflavin(mg) | 0.3 | 0.7 | 1.2 | <0.001 | 0.9 | 0.5 | 0.5 | <0.001 | 0.7 | 0.5 | 0.8 | <0.001 |
| Folate(μg) | 141 | 273 | 510 | <0.001 | 369 | 230 | 238 | <0.001 | 315 | 225 | 293 | <0.001 |
| Niacin(mg) | 7 | 11 | 18 | <0.001 | 14 | 10 | 10 | <0.001 | 13 | 10 | 11 | <0.001 |
| Vitamin C(mg) | 42 | 89 | 166 | <0.001 | 97 | 73 | 98 | <0.001 | 120 | 73 | 77 | <0.001 |
| Vitamin E(mg) | 6.9 | 16.6 | 31 | <0.001 | 22 | 14 | 14 | <0.001 | 19 | 14 | 17 | <0.001 |
| Ca(mg) | 235 | 462 | 795 | <0.001 | 666 | 392 | 362 | <0.001 | 470 | 392 | 545 | <0.001 |
| Fe(mg) | 8.3 | 14.6 | 25.8 | <0.001 | 20 | 13 | 13 | <0.001 | 17 | 13 | 16 | <0.001 |
| Zn(mg) | 2.8 | 5.3 | 9.6 | <0.001 | 7.3 | 4.4 | 4.6 | <0.001 | 5.8 | 4.5 | 6.1 | <0.001 |
| Se(μg) | 17 | 28 | 49 | <0.001 | 40 | 24 | 22 | <0.001 | 27 | 24 | 34 | <0.001 |

T, tertiles; RE, retinol equivalent.

* P values for the differences among the groups of dietary pattern scores were derived from the $\chi^2$ test for categorical variables and the Kruskal–Wallis test for continuous variables.

† Values are presented as percentages for these categorical variables.

Likewise, there were lower prevalence of SGA in T3 group of equilibrium pattern and T1 group of snacks pattern (P<0.05).

## Association between dietary patterns and abnormal birth weight using non-spatial logistic regression

Non-spatial logistic regression results showed that SGA was less likely to occur among the infants born by mothers in the T3 group of equilibrium pattern in comparison with those whose mothers were from T2 group (OR: 0.801, 95%CI: 0.665–0.965) (Table 4). Mothers with higher scores of snacks pattern were associated with increasing risk of macrosomia and SGA among the infants (OR: 1.265, 95%CI: 1.000–1.602; OR: 1.260, 95%CI: 1.056–1.505, respectively). Compared with those in the T2 group of prudent pattern, mothers in the T1 group of the prudent pattern had higher risk of LBW in their offspring (OR: 0.732, 95%CI: 0.537–0.997), and mothers in the T3 group had lower risk of fetal macrosomia (OR: 0.788, 95%CI: 0.623–0.997).

## Association between dietary patterns and abnormal birth weight using spatial logistic regression

Based on geographically weighted regression (GWR) for LBW, macrosomia, SGA and LGA in the present study, the optimal bandwidth size were 7332.000, 6631.000, 7596.000, and 7573.000 (persons) respectively. Comparing the global regression and GWR, we concluded that the local regression has better fitness. Approximately 81.6% of mothers with higher scores on the equilibrium pattern was more unlikely to have SGA infants (P < 0.05), with the highest

**Table 3. Baseline characteristics of study population (n 7934) in Shaanxi province, Northwest China, 2013[*].**

| Study variable | LBW | Macrosomia | SGA | LGA |
| --- | --- | --- | --- | --- |
| | n (%) | n (%) | n (%) | n (%) |
| Child gender | | | | |
| Male | 139(3.3) | 316(7.5) | 461(11.0) | 436(10.4) |
| Female | 157(4.3)[†] | 174(4.8)[†] | 407(11.3) | 358(9.9) |
| Fetal number | | | | |
| Singleton | 251(3.2) | 488(6.3) | 827(10.7) | 786(10.2) |
| Twin and multi-fetal | 45(46.9)[†] | 2(2.1) | 41(43.2)[†] | 8(8.4) |
| Infant parity | | | | |
| 1 | 148(3.2) | 251(5.4) | 500(11.0) | 414(9.1) |
| 2 | 120(4.1) | 220(7.5) | 318(10.9) | 350(11.9) |
| ≥3 | 26(10.3)[†] | 18(7.1)[†] | 42(16.7)[†] | 28(11.1)[†] |
| Childbearing age | | | | |
| 18–24 | 95(3.5) | 134(5.0) | 336(12.6) | 216(8.1) |
| 25–29 | 126(3.9) | 200(6.2) | 328(10.3) | 324(10.1) |
| ≥30 | 74(3.9) | 152(8.1)[†] | 200(10.8)[†] | 243(13.1)[†] |
| Mother's education | | | | |
| Primary school and below | 27(3.8) | 46(6.4) | 92(13.0) | 67(9.4) |
| Junior high school | 173(4.1) | 242(5.7) | 503(12.0) | 416(9.9) |
| Senior high school | 71(4.1) | 110(6.3) | 185(10.8) | 165(9.6) |
| College and above | 25(2.1)[†] | 91(7.7) | 87(7.5)[†] | 142(12.3) |
| Mother's residence during pregnancy | | | | |
| Floating | 43(3.7) | 64(6.2) | 106(11.2) | 111(10.1) |
| Permanent | 252(4.3) | 423(6.4) | 755(10.7) | 678(11.2) |
| Household wealth Index | | | | |
| Poor | 104(4.1) | 133(5.2) | 329(13.0) | 221(8.8) |
| Middle-income | 89(3.4) | 167(6.4) | 274(10.6) | 264(10.2) |
| Rich | 103(3.8) | 190(7.0)[†] | 265(9.9)[†] | 309(11.5)[†] |
| Altitude | | | | |
| <500 | 74(3.2) | 164(7.0) | 198(8.6) | 250(10.8) |
| 500–1000 | 142(3.9) | 221(6.1) | 396(11.1) | 372(10.5) |
| >1000 | 80(4.1) | 105(5.4) | 274(14.3)[†] | 172(9.0) |
| Area | | | | |
| South area | 100(4.1) | 148(6.1) | 258(10.9) | 244(10.3) |
| Central area | 148(3.7) | 248(6.2) | 467(11.7) | 406(10.2) |
| North area | 48(3.3) | 94(6.6) | 143(10.0) | 144(10.1) |
| Equilibrium pattern | | | | |
| T1 | 116(4.4) | 163(6.2) | 310(12.0) | 247(9.5) |
| T2 | 102(3.9) | 148(5.6) | 312(12.0) | 262(10.1) |
| T3 | 78(3.0)[†] | 179(6.8) | 246(9.5)[†] | 285(11.0) |
| Snacks pattern | | | | |
| T1 | 76(2.9) | 171(6.5) | 254(9.8) | 289(11.1) |
| T2 | 105(4.0) | 143(5.4) | 284(10.9) | 241(9.3) |
| T3 | 115(4.4)[†] | 176(6.7) | 330(12.7)[†] | 264(10.2) |
| Prudent pattern | | | | |
| T1 | 87(3.3) | 157(6.0) | 296(11.4) | 247(9.5) |
| T2 | 121(4.6) | 176(6.7) | 299(11.5) | 277(10.7) |

(*Continued*)

**Table 3.** (Continued)

| Study variable | LBW | Macrosomia | SGA | LGA |
|---|---|---|---|---|
| | n (%) | n (%) | n (%) | n (%) |
| T3 | 88(3.3)[†] | 157(6.0) | 273(10.5) | 270(10.4) |

T, tertiles; LBW, low birth weight; SGA, small for gestational age; LGA, large for gestational age.

[*] Values are the number of abnormal birth outcomes and the prevalence is included in the bracket.

[†] Significant differences among the groups were derived from the $\chi^2$ test for categorical variables (P<0.05).

OR observed in North Shaanxi and the lowest OR observed in South Shaanxi. (Table 5 and S5 Fig). The higher scores of mothers with snacks pattern during pregnancy were found to be positively associated with LBW for 23.2% of participants (P< 0.05), with decreasing OR values from central Shaanxi to the north and south Shaanxi (Table 5 and S6 Fig). Similarly, infants born by women in the T3 group of snacks pattern was significantly correlated with macrosomia for 19.5% of participants (P< 0.05), with decreasing OR values from south to north (Table 5 and S7 Fig). Among about 80.0% of participants who lived in South and Central Shaanxi, there was a higher risk for SGA in the T3 group of snacks pattern than that in the T2 group (P < 0.05). The OR values tend to descend from South to North Shaanxi (Table 5 and

**Table 4. The association of diet pattern with abnormal birth outcomes using non-spatial logistical regression[*].**

| | Equilibrium pattern | | | Snacks pattern | | | Prudent pattern | | |
|---|---|---|---|---|---|---|---|---|---|
| | T1 | T2 | T3 | T1 | T2 | T3 | T1 | T2 | T3 |
| **LBW** | | | | | | | | | |
| Crude OR[a] | 1.032 | 1 | 0.797 | 0.786 | 1 | 1.154 | 0.726 | 1 | 0.793 |
| | (0.780–1.366) | | (0.587–1.081) | (0.569–1.085) | | (0.878–1.516) | (0.544–0.969) | | (0.595–1.058) |
| Adjusted OR[b] | 1.075 | 1 | 0.844 | 0.783 | 1 | 1.226 | 0.732 | 1 | 0.788 |
| | (0.800–1.444) | | (0.612–1.164) | (0.557–1.100) | | (0.919–1.635) | (0.537–0.997) | | (0.581–1.069) |
| **Macrosomia** | | | | | | | | | |
| Crude OR[a] | 1.108 | 1 | 1.205 | 1.207 | 1 | 1.237 | 0.874 | 1 | 0.846 |
| | (0.875–1.403) | | (0.959–1.515) | (0.942–1.546) | | (0.983–1.557) | (0.695–1.098) | | (0.673–1.063) |
| Adjusted OR[b] | 1.137 | 1 | 1.169 | 1.183 | 1 | 1.265 | 0.853 | 1 | 0.788 |
| | (0.894–1.446) | | (0.925–1.479) | (0.920–1.523) | | (1.000–1.602) | (0.674–1.080) | | (0.623–0.997) |
| **SGA** | | | | | | | | | |
| Crude OR[a] | 0.969 | 1 | 0.769 | 0.962 | 1 | 1.236 | 1.006 | 1 | 0.956 |
| | (0.816–1.152) | | (0.642–0.920) | (0.793–1.166) | | (1.042–1.465) | (0.844–1.199) | | (0.799–1.143) |
| Adjusted OR[b] | 0.968 | 1 | 0.801 | 0.996 | 1 | 1.26 | 0.997 | 1 | 0.996 |
| | (0.811–1.157) | | (0.665–0.965) | (0.817–1.215) | | (1.056–1.505) | (0.829–1.198) | | (0.828–1.199) |
| **LGA** | | | | | | | | | |
| Crude OR[a] | 0.946 | 1 | 1.087 | 1.172 | 1 | 1.097 | 0.856 | 1 | 0.92 |
| | (0.783–1.142) | | (0.907–1.302) | (0.964–1.425) | | (0.911–1.321) | (0.711–1.030) | | (0.766–1.103) |
| Adjusted OR[b] | 0.948 | 1 | 1.069 | 1.124 | 1 | 1.102 | 0.836 | 1 | 0.88 |
| | (0.782–1.149) | | (0.888–1.287) | (0.921–1.373) | | (0.911–1.334) | (0.690–1.013) | | (0.730–1.062) |

T, tertiles; LBW, low birth weight; SGA, small for gestational age; LGA, large for gestational age.

[*] Values are OR of abnormal birth outcomes and its 95% confidence interval is included in the bracket.

[a] OR are not adjusted for any variables.

[b] OR are adjusted for socio-demographic characteristics (child gender, fetal number, infant parity, childbearing age, mother's education, mother's residence during pregnancy, Household wealth Index, altitude of residence and area).

**Table 5. The association of dietary pattern with abnormal birth outcomes using the GWR model*.**

| | Equilibrium pattern | | | Snacks pattern | | | Prudent pattern | | |
|---|---|---|---|---|---|---|---|---|---|
| | T1 | T2 | T3 | T1 | T2 | T3 | T1 | T2 | T3 |
| *LBW* | | | | | | | | | |
| Crude OR[a] | 0.993(0.895–1.125) | 1 | 0.889(0.742–0.998) | 0.886(0.778–0.946) | 1 | 1.126(1.037–1.249) | 0.825(0.797–0.919) | 1 | 0.807(0.776–0.967) |
| Adjusted OR[b] | 1.037(0.975–1.135) | 1 | 0.942(0.796–1.007) | 0.876(0.819–0.953) | 1 | 1.145(1.066–1.247) | 0.814(0.766–0.892) | 1 | 0.835(0.779–0.951) |
| Percentage[†] | 0 | | 0 | 6.2 | | 23.2 | 75.0 | | 59.3 |
| *Macrosomia* | | | | | | | | | |
| Crude OR[a] | 1.040(1.029–1.065) | 1 | 1.104(1.073–1.143) | 1.080(1.019–1.132) | 1 | 1.099(1.009–1.162) | 0.934(0.904–0.950) | 1 | 0.940(0.921–0.965) |
| Adjusted OR[b] | 1.045(0.985–1.089) | 1 | 1.099(1.037–1.149) | 1.017(0.969–1.161) | 1 | 1.076(1.027–1.212) | 0.929(0.889–0.957) | 1 | 0.912(0.875–0.939) |
| Percentage[†] | 0 | | 0 | 6.6 | | 19.5 | 0 | | 19.2 |
| *SGA* | | | | | | | | | |
| Crude OR[a] | 0.997(0.827–1.103) | 1 | 0.836(0.740–1.073) | 0.983(0.878–1.099) | 1 | 1.150(0.936–1.235) | 1.045(0.962–1.088) | 1 | 0.960(0.875–1.192) |
| Adjusted OR[b] | 0.989(0.972–1.053) | 1 | 0.890(0.883–0.985) | 1.007(0.996–1.015) | 1 | 1.104(1.048–1.112) | 0.978(0.970–0.985) | 1 | 0.995(0.984–1.004) |
| Percentage[†] | 0 | | 81.6 | 0 | | 80.0 | 0 | | 0 |
| *LGA* | | | | | | | | | |
| Crude OR[a] | 0.988(0.927–0.998) | 1 | 1.064(1.035–1.102) | 1.092(0.981–1.120) | 1 | 1.052(0.973–1.078) | 0.924(0.908–0.962) | 1 | 0.958(0.954–0.987) |
| Adjusted OR[b] | 0.985(0.932–0.999) | 1 | 1.053(1.015–1.097) | 1.070(0.962–1.097) | 1 | 1.057(0.988–1.079) | 0.927(0.914–0.933) | 1 | 0.934(0.931–0.959) |
| Percentage[†] | 0 | | 0 | 0 | | 0 | 0 | | 0 |

T, tertiles; LBW, low birth weight; SGA, small for gestational age; LGA, large for gestational age.

* Values are middle OR of abnormal birth outcomes and its maximum and minimum are included in the bracket.

[a] OR are not adjusted for any variables.

[b] OR are adjusted for socio-demographic characteristics (child gender, fetal number, infant parity, childbearing age, mother's education, mother's residence during pregnancy, Household wealth Index and altitude of residence).

[†] Percentage of the sample that was statistically significant after adjusting for socio-demographic characteristics.

S8 Fig). With T2 group of prudent pattern as the reference, a lower risk for LBW among the infants were observed among approximately 75.0% of participants in T1 group and 59.3% of those in T3 group, respectively (P< 0.05). For the OR values of LBW, there was a tendency to decrease from South to North Shaanxi (Table 5, S9 and S10 Figs). The mothers with higher scores of prudent pattern was also negatively associated with the increasing risk of fetal macrosomia among 19.2% of participants (P<0.05), with adjusted OR values ranging from 0.875 (South Shaanxi) to 0.939 (Central Shaanxi) (Table 5 and S11 Fig).

## Discussion

In the present study, three dietary patterns (equilibrium pattern, prudent pattern and snacks pattern) were identified among the mothers during pregnancy in Shaanxi province of northwestern China. We used spatial regression models to explore the geographic variation of maternal dietary patterns in relation to the risk of abnormal birth weight after controlling for other baseline characteristics.

Previous studies suggested that the single nutrient analysis in nutritional epidemiology failed to account for interactions between food components or foods. Dietary patterns represent a broader picture of food and nutrient intakes and are able to account for cumulative and interactive effects [5]. The results suggested that pregnant women with high adherence to the equilibrium pattern tended to eat a diet with a balance of all of the food groups and foods that are rich in energy and various nutrients. Moreover, participants with high adherence to the equilibrium pattern were more likely to be better educated, wealthier, 25–29 years old at

delivery, and living at lower altitudes. The characteristics of this dietary model identified in our study was homologous to the dietary patterns in Guangzhou, China [24]. The prudent pattern was characterized by high intakes of dairy products and eggs, and the snacks pattern had high loading of beverages, sweetmeat, and fast-food. Those with higher scores on the prudent pattern and the snacks pattern had lower energy and carbohydrate intake and lower intakes of some selected nutrients. The sociodemographic distribution of mothers with prudent pattern was similar to those with equilibrium pattern, while those with high scores on the snacks pattern were more likely to live in areas with higher altitude (>1000 meters), be poor and less educated, and be aged 18–24 years at delivery. Some studies reported that higher education level and better economic status might enable individuals to make healthy diet choices [25, 26]. Hence, people with higher education level and better economic status tend to be more inclined to adopt the equilibrium pattern [27–30]. Conversely, low socioeconomic groups with relatively low level of nutrition knowledge may make unhealthy diet choices and lead to a poor-quality snacks pattern [31].

Due to the huge difference in geographical environment and socio-culture across various areas in China, it was well recognized that the dietary habits and food consumption have different geographical distribution [32, 33]. Similarly, dietary habits also varied largely among different areas in Shaanxi [34]. In our studies, the highest scores of the equilibrium pattern were observed in Central Shaanxi, while the lowest score of the equilibrium pattern was observed in North Shaanxi. The scores of the snacks pattern were higher in North Shaanxi, followed by South Shaanxi and Central Shaanxi. It appeared a decreasing trend of the scores of the prudent pattern from south to north. It is possible that the different geographical environment (e.g. the area altitudes) resulted in different geographical distribution of dietary patterns.

Healthy diet characterized by consuming a variety of nutritious foods during pregnancy is essential for promoting and maintaining fetal health [7, 28]. Global regression analysis revealed that mothers with high adherence to the equilibrium pattern during pregnancy were more unlikely to give birth to a baby with SGA. The findings suggested that equilibrium pattern with a balance of all of the food groups and foods may be beneficial to reduce the risk of SGA, which provided more evidence that healthy diets promote birth weight. Further, the GWLR model showed that equilibrium pattern might reduce more neonates with SGA in Central and South Shaanxi than those in North Shaanxi. Our survey data indicated that the scores of equilibrium pattern was higher in Central and South Shaanxi than those in North Shaanxi. Most parts of Central Shaanxi are developed urban areas (e.g., Xi 'an) where mothers have higher household economic levels and higher education. In contrast, the surveyed area in North Shaanxi comprises of small townships and villages where mothers have lower economic levels and are less educated. Pregnant women in villages with greater density are more likely to eat high-starch foods and tend to adopt more frugal dietary structure [35]. These results remind us that it is extremely urgent to advocate the equilibrium diet during pregnancy, especially in North Shaanxi. Additional studies will be required to corroborate these findings and to elucidate possible reasons for this geographical variations.

There is a high intake of snack food among the Chinese in recent years [36]. The snacks foods as an unhealthy food choice is unable to fulfill the increased key micronutrients requirements for normal fetal growth and may lead to a higher risk of some diseases [37]. According to the global regression, the higher scores of snacks pattern was associated with the increasing risk of macrosomia and SGA among the infants. The snacks pattern consists of foods with high concentrations of simple carbohydrates, lipids and low amounts of protein and micronutrients. Insufficient protein intake during pregnancy affects fetal organ development, leading to a variety of adverse pregnancy outcomes [38]. During pregnancy, fetal growth and development are very rapid, and inadequate nutrition intake can lead to fetal growth retardation and

low birth weight [39]. However, some studies also found that this combination of substances in the snacks pattern may lead to rapid fetal weight gain and increase the incidence of macrosomia [40]. Therefore, the snacks pattern contains unbalanced nutrients, favoring abnormal birth weight [41, 42].

GWLR model further revealed that snacks pattern during pregnancy may increase the risk of LBW, Macrosomia and SGA in 23.2%, 19.5% and 80.0%, respectively of all participants. When we focus on the map drawn by OR values, it is clear that the highest OR values of LBW were in Central Shaanxi, and OR values of Macrosomia and SGA progressively decreased from South to North Shaanxi. Thus, it is essential to improve the maternal diet status during pregnancy according to regional characteristics.

The spatial regression model revealed a negative association between high propensity of prudent pattern and the risk of LBW in 59.3% of mothers residing in Central Shaanxi and parts of North Shaanxi. On the map used to visualize these results, prudent pattern might decrease more neonates with LBW in North Shaanxi, followed by Central Shaanxi. The negative association of prudent pattern with macrosomia were mainly present in South Shaanxi. Because the prudent pattern has high contents of dairy products and eggs, which included higher protein and micronutrients compared to snacks pattern, it might benefit maintaining appropriate birth weight. However, the geographical variations after controlling for the important risk factors provide potential policy implications, calling for further investigations.

However, we have to acknowledge some limitations that need to be addressed in the study. First, due to the nature of cross-sectional design, we cannot draw any cause-effect conclusion based on the study results. Secondly, it was difficult to isolate the variation trends in relevant factors based on limited number of areas sampled in our study. Thirdly, we adopted the mothers' self-reported information on dietary intake during pregnancy. Thus, the possibility of information bias should be considered. Different degree of the reminder bias based on different birth dates is also one of the limitations of our survey even though we adopted systematic approaches to maximize power and minimize bias. Fourthly, the study may be subject to some potential confounders. For example, maternal nutrition status of pregnant (BMI), drug use, genetics, and environmental factors were not available and thus could not be controlled in multivariate analysis. Despite these limitations, this study has still provided important information on the geographic varying association of dietary patterns with abnormal birth weight in northwestern China.

## Conclusions

In summary, three dietary patterns (equilibrium pattern, prudent pattern and snacks pattern) were extracted among the mothers during pregnancy, which had different geographical distribution in Shaanxi province of northwestern China. The equilibrium pattern, which is rich in energy and various nutrients, was negatively associated with SGA. The prudent pattern were more likely to decrease the risk of LBW and macrosomia, respectively mainly in North Shaanxi and South Shaanxi. The snacks pattern during pregnancy may cause abnormal birth weight (e.g. LBW, macrosomia, SGA) among infants, especially in Central and South Shaanxi. The findings emphasize the importance of geographical distribution to improve the dietary patterns among disadvantaged pregnant women.

## Supporting information

**S1 Table. The association of diet pattern with abnormal birth outcomes using non-spatial logistical regression in South Shaanxi.**
(DOCX)

**S2 Table. The association of diet pattern with abnormal birth outcomes using non-spatial logistical regression in Central Shaanxi.**
(DOCX)

**S3 Table. The association of diet pattern with abnormal birth outcomes using non-spatial logistical regression in North Shaanxi.**
(DOCX)

**S4 Table. The characteristics of dietary patterns among pregnant women (n 7934) in Shaanxi province, Northwest China, 2013.**
(DOCX)

**S5 Table. The association of diet pattern with abnormal birth outcomes using non-spatial logistical regression.**
(DOCX)

**S1 Fig. Geographical distribution of the incidence of LBW among the live infants in Shaanxi province, 2013.**
(TIF)

**S2 Fig. Geographical distribution of the incidence of macrosomia among the live infants in Shaanxi province, 2013.**
(TIF)

**S3 Fig. Geographical distribution of the incidence of SGA among the live infants in Shaanxi province, 2013.**
(TIF)

**S4 Fig. Geographical distribution of the incidence of LGA among the live infants in Shaanxi province, 2013.**
(TIF)

**S5 Fig. Geographical distribution of adjusted ORs of equilibrium pattern of T3, as associated with prevalence of SGA.**
(TIF)

**S6 Fig. Geographical distribution of adjusted ORs of snacks pattern of T3, as associated with prevalence of LBW.**
(TIF)

**S7 Fig. Geographical distribution of adjusted ORs of snacks pattern of T3, as associated with prevalence of macrosomia.**
(TIF)

**S8 Fig. Geographical distribution of adjusted ORs of snacks pattern of T3, as associated with prevalence of SGA.**
(TIF)

**S9 Fig. Geographical distribution of adjusted ORs of prudent pattern of T1, as associated with prevalence of LBW.**
(TIF)

**S10 Fig. Geographical distribution of adjusted ORs of prudent pattern of T3, as associated with prevalence of LBW.**
(TIF)

**S11 Fig. Geographical distribution of adjusted ORs of prudent pattern of T3, as associated with prevalence of macrosomia.**
(TIF)

**S1 Data.**
(SAV)

# Acknowledgments

We thank the women and their children participated in our study. We thank local hospitals and health bureau as well as the Ministry of Health in Shaanxi province.

# Author Contributions

**Conceptualization:** Yini Liu, Leilei Pei.

**Data curation:** Yini Liu, Huihui Zhang, Yaling Zhao, Fangyao Chen, Baibing Mi.

**Formal analysis:** Yini Liu, Yaling Zhao, Fangyao Chen, Leilei Pei.

**Funding acquisition:** Leilei Pei.

**Investigation:** Yini Liu, Huihui Zhang, Jing Zhou, Yulong Chen.

**Methodology:** Yini Liu, Duolao Wang, Leilei Pei.

**Project administration:** Leilei Pei.

**Software:** Yini Liu, Baibing Mi, Leilei Pei.

**Supervision:** Duolao Wang, Leilei Pei.

**Writing – original draft:** Yini Liu.

**Writing – review & editing:** Yini Liu, Leilei Pei.

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
