## [Decision Letter · Decision Letter 0]

9 Feb 2021

PONE-D-20-40026

Geographical variations in maternal dietary patterns during pregnancy associated with birth weight in Shaanxi province, Northwestern China

PLOS ONE

Dear Dr. Pei,

Thank you for submitting your manuscript to PLOS ONE. After careful consideration, we feel that it has merit but does not fully meet PLOS ONE’s publication criteria as it currently stands. Therefore, we invite you to submit a revised version of the manuscript that addresses the points raised during the review process.

An expert in the field handled your manuscript, and we are very appreciative of their time and contributions. Although some interest was found in your study, several major comments arose. Please address ALL of the reviewer's comments in your revised manuscript.

We look forward to receiving your revised manuscript.

Kind regards,

Frank T. Spradley

Academic Editor

PLOS ONE

3. Please include additional information regarding the survey or questionnaire used in the study and ensure that you have provided sufficient details that others could replicate the analyses. For instance, if you developed a questionnaire as part of this study and it is not under a copyright more restrictive than CC-BY, please include a copy, in both the original language and English, as Supporting Information, or include a citation if it has been published previously.

4. In statistical methods, please refer to any post-hoc corrections to correct for multiple comparisons during your statistical analyses. If these were not performed please justify the reasons. Please refer to our statistical reporting guidelines for assistance (https://journals.plos.org/plosone/s/submission-guidelines.#loc-statistical-reporting).

6. We note that Figures 1-3 and Supporting Figures 1-11 in your submission contain map images which may be copyrighted. All PLOS content is published under the Creative Commons Attribution License (CC BY 4.0), which means that the manuscript, images, and Supporting Information files will be freely available online, and any third party is permitted to access, download, copy, distribute, and use these materials in any way, even commercially, with proper attribution. For these reasons, we cannot publish previously copyrighted maps or satellite images created using proprietary data, such as Google software (Google Maps, Street View, and Earth). For more information, see our copyright guidelines: http://journals.plos.org/plosone/s/licenses-and-copyright.

6.1.    You may seek permission from the original copyright holder of Figures 1-3 and Supporting Figures 1-11 to publish the content specifically under the CC BY 4.0 license. 

6.2.    If you are unable to obtain permission from the original copyright holder to publish these figures under the CC BY 4.0 license or if the copyright holder’s requirements are incompatible with the CC BY 4.0 license, please either i) remove the figure or ii) supply a replacement figure that complies with the CC BY 4.0 license. Please check copyright information on all replacement figures and update the figure caption with source information. If applicable, please specify in the figure caption text when a figure is similar but not identical to the original image and is therefore for illustrative purposes only.

7. Thank you for submitting the above manuscript to PLOS ONE. During our internal evaluation of the manuscript, we found significant text overlap between your submission and the following previously published works, some of which you are an author.

- https://www.nature.com/articles/s41598-020-69788-0

- https://www.cambridge.org/core/services/aop-cambridge-core/content/view/339BB3F814A892B08496A19E889B405A/S1368980016002159a.pdf/dietary_intakes_and_dietary_patterns_among_pregnant_women_in_northwest_china.pdf

- https://pubmed.ncbi.nlm.nih.gov/32737435/

- https://www.sciencedirect.com/science/article/abs/pii/S0304389420303654?via%3Dihub

- https://www.nature.com/articles/s41598-017-07076-0?code=a32ec384-3b72-4a76-bf86-568a96975613&error=cookies_not_supported

Please revise the manuscript to rephrase the duplicated text, cite your sources, and provide details as to how the current manuscript advances on previous work. Please note that further consideration is dependent on the submission of a manuscript that addresses these concerns about the overlap in text with published work.

Reviewers' comments:

Reviewer's Responses to Questions

**Comments to the Author**

1. Is the manuscript technically sound, and do the data support the conclusions?

Reviewer #1: Partly

2. Has the statistical analysis been performed appropriately and rigorously? 

Reviewer #1: N/A

3. Have the authors made all data underlying the findings in their manuscript fully available?

Reviewer #1: No

4. Is the manuscript presented in an intelligible fashion and written in standard English?

Reviewer #1: Yes

5. Review Comments to the Author

Reviewer #1: This is a fascinating manuscript. However, confirmation must be added to provide better detail.

Please consider adding confirmation to respond to a question or consider revising the manuscript following the file review below.

A.1. Line 81. The survey was held from August to November 2013. Line 82, the objects were live birth born during 2010-2013. Kindly add specific time as inclusion criteria of this study to describe the study object more clearly e.g Baby born in January-August 2013 . Add in the method section.

A.2. Line 82. Material and methods. The diet survey questionnaire was used on the basis of the respondent's ability to recall history during pregnancy. Certainly, it has potential reminder bias.The subjects in this analysis are live baby mothers born in 2010-2013. It indicates different age ranges.

How does the author control the reminder bias based on different birth dates, e.g., the neonate mother (0-28 days) can remember better a dietary frequency than moms between 12-44 months of age? Perhaps you can explain the method of quality control.

A.3. Line 89. Why did the author make a different number of babies born as a topic in a rural and urban area? Ten villages and 20 communities? Depending on the number of children born in both areas? Has the author found a list of babies born in every county in urban or rural areas? Are the samples selected by the proportionate probability size (PPS) method? If yes. Please add in the process.If not, does the author use a weight to statistical analysis?

A.4. Line 130. Did the pattern equilibrium, snack, and prudent was derived from the explanatory factor/ confirmatory factor? Before taking PCA analysis? Add in the method.

What software was used to doing PCA in this study? SPSS? STATA? Describe in this section.

A.5. If available, please identify geographical areas and add on the results section too? It is essential, the readers know about the classification, maybe most of the geographical kind, of three classified areas (northern, central, and southern), which is the coastal area? Mountainous? High land? For Urban, it is metropolitan, small-town Probably it related to different dietary trends

A.6. Line 175. Most of the respondents live in central province Shaanxi. Is the distribution of participant influence in the dietary pattern? Maybe potential to be modified factor. Please explain how the author took a sample from this large survey for this study purposes in the Methods section.

Why did the author consider using the whole data of a large survey? Did the author consider resampling to this analysis purposes so it can representative in three areas? Due to the aim study to comparing three area?

A.7. Is the author want to compare three areas? The author shows the distribution of dietary patterns based on geographical area, while the author also shows the participant proportion based on three areas is different (unbalanced). The various distribution proportions on the three areas potentially be a modifying factor. In table 2, we can see a significantly different dietary pattern between the three areas based on the X2 test. How the author control this issue?

A.8. Table 2. Kindly show the number of samples n (%) to shows balanced samples in each category.

A.9. Line 174 shows the different distribution of study participants. Has the author previously checked the various respondents' distribution in this area, probably influence this significance? Potential to be modified factor due to unbalanced distribution of samples, or consider reporting the effect in each region

(stratification) in multivariate if the study investigates the association between risk factor of birth outcome.

A.10. Table 2 and 3. For association purposes, the author can include adjusted p value from the multivariate analysis in this table with different mark, e.g ††

A.11. Line 220. We suggest not to use “varied in line 220“ because the proportion shows a slight difference.

A.12. Table 3. Refers to comment A10, to add p value multivariat analysis with different mark ††

A.13. Table 4. There is no socio-demographic characteristic was shown in this table, is only the dietary pattern that was a significant association with the birth outcome ? kindly add in the complete information in the previous table for socio demographics characteristic for multivariate test

A.14. General comment for table 5. How is it far different using nonspatial logistic regression and GWR?

A.15. Line 307 . Does it mean the different patterns mostly influenced by altitude? Why didn’t the author show others based on the rural and urban type of area

A.16. Line 354. Birth weight outcomes have close relation with the BMI of the mother (nutrition status of pregnant). Did the author consider this variable in their analysis or take in the limitation study.

6. PLOS authors have the option to publish the peer review history of their article (what does this mean?). If published, this will include your full peer review and any attached files.

Reviewer #1: No

---

## [Author Response · Author response to Decision Letter 0]

21 Jun 2021

A.1. Line 81. The survey was held from August to November 2013. Line 82, the objects were live birth born during 2010-2013. Kindly add specific time as inclusion criteria of this study to describe the study object more clearly e.g Baby born in January- August 2013. Add in the method section.

Reply: We are thankful for you to point out the detailed problem. The specific content has been revised in the method section.

A.2. Line 82. Material and methods. The diet survey questionnaire was used on the basis of the respondent's ability to recall history during pregnancy. Certainly, it has potential reminder bias.The subjects in this analysis are live baby mothers born in 2010-2013. It indicates different age ranges. 

How does the author control the reminder bias based on different birth dates, e.g., the neonate mother (0-28 days) can remember better a dietary frequency than moms between 12-44 months of age? Perhaps you can explain the method of quality control.

Reply: Thanks very much for your question. Because the reported information was self-reported by mothers, the duration of time that had elapsed between pregnancy and the maternal self-report may have contributed to recall bias. Different degree of reminder bias based on different birth dates is also one of the limitations of our cross-sectional survey. Actually, this limitation should be considered while interpreting our results. In terms of quality control, all questionnaires and procedures were pre-tested, and detailed interviewer guides were developed before the formal survey. We also consciously adopted systematic approaches to maximize power and minimize bias in the study.

A.3. Line 89. Why did the author make a different number of babies born as a topic in a rural and urban area? Ten villages and 20 communities? Depending on the number of children born in both areas? Has the author found a list of babies born in every county in urban or rural areas? Are the samples selected by the proportionate probability size (PPS) method? If yes. Please add in the process. If not, does the author use a weight to statistical analysis?

Reply: Thanks for pointing out the question. For the current study, the samples were selected by the proportionate probability size (PPS) method, according to the proportion of urban and rural areas, and considering the population density and fertility level. The differences between urban and rural areas in this sample were mainly determined according to the total population of urban and rural areas in the whole province so that the sample was representative. 

Actually, the study data had been conducted a series of researches and published many articles. As follows:

[1]. Pei, L. L., Kang, Y. J., Cheng, Y. &Yan, H. The Association of Maternal Lifestyle with Birth Defects in Shaanxi Province, Northwest China. PLoS One.2015. 10: e0139452.

[2]. Pei, L. L., Kang, Y. J., Zhao, Y. L. &Yan, H. Prevalence and risk factors of congenital heart defects among live births: a population-based cross-sectional survey in Shaanxi province, Northwestern China. BMC Pediatrics.2017. 17: 18.

[3]. Liu Y, Zhang H, Li J, et al. Geographical variations in maternal lifestyles during pregnancy associated with congenital heart defects among live births in Shaanxi province, Northwestern China[J]. Sci Rep. 2020, 10(1): 12958.

A.4. Line 130. Did the pattern equilibrium, snack, and prudent was derived from the explanatory factor/ confirmatory factor? Before taking PCA analysis? Add in the method. What software was used to? SPSS? STATA? Describe in this section.

Reply: We should apologize for the avoidable errors in the manuscript. The equilibrium, snacks, and prudent pattern were extracted by principal component analysis (PCA), not factor analysis, which was an expression error. Meanwhile, SPSS software was used to doing PCA in this study. We have corrected the errors in the original manuscript in the methods section.

5. A.5. If available, please identify geographical areas and add on the results section too? It is essential, the readers know about the classification, maybe most of the geographical kind, of three classified areas (northern, central, and southern), which is the coastal area? Mountainous? High land? For Urban, it is metropolitan, small-town. Probably it related to different dietary trends.

Reply: First of all, thanks for your valuable opinion. Shaanxi is a landlocked province in Northwest China and spans a wide latitude. From south to north, Shaanxi is divided into three physiographic regions by the North Mountains and the Qin Mountains. These three parts are North Shaanxi, Central Shaanxi, and South Shaanxi. North Shaanxi is a plateau, 900-1500 meters above sea level. The winter in North Shaanxi is chilly, and the difference in temperature between day and night is big. Central Shaanxi is a plain, 520 meters above sea level. The summer is very hot, and the winter is dry and cold. South Shaanxi is a mountainous region with an elevation of about 1500-3000 meters. Summer is the rainy period in South Shaanxi, and winter is chilly. Shaanxi’s agricultural production varies throughout the province. Shaanxi’s northern plateau grows spring-sown wheat and millet. Central Shaanxi is an important producing area of wheat and corn in north China. South Shaanxi is the main producing area of rice and cole. There are also significant differences in the economic conditions of different regions，the specific situation is as follows. According to data from the Shaanxi Provincial Bureau of Statistics in 2011, per capita GDP in northern Shaanxi was 61,402 yuan, that in Central Shaanxi was 32,892 yuan, and that in South Shaanxi was 16,912 yuan.

https://cn.bing.com/search?q=Shaanxi&qs=n&form=QBRE&sp=-1&pq=shaanxi&sc=8-7&sk=&cvid=D9A6A35C5B024905914905ADF5A5D420

A.6. Line 175. Most of the respondents live in central province Shaanxi. Is the distribution of participant influence in the dietary pattern? Maybe potential to be modified factor. Please explain how the author took a sample from this large survey for this study purposes in the Methods section. Why did the author consider using the whole data of a large survey? Did the author consider resampling to this analysis purposes so it can representative in three areas? Due to the aim study to comparing three area?

Reply: I am very grateful for your comments on the manuscript. According to the Sixth national population census data of Shaanxi Province in 2010 released by the National Bureau of Statistics, the population ratios of South Shaanxi, Central Shaanxi, and North Shaanxi were 22.5%, 62.7%, and 14.8% respectively. It can be seen that the population distribution of different areas in Shaanxi Province was originally unbalanced. Meanwhile, the proportion of births in 2011 was 22.4% in South Shaanxi, 61.2% in Central Shaanxi, and 16.4% in North Shaanxi according to the Shaanxi Provincial Bureau of Statistics. Therefore, the samples investigated in this study are representative to some extent in the three areas. Detailed sampling methods were in the Methods section.

A.7. Is the author want to compare three areas? The author shows the distribution of dietary patterns based on geographical area, while the author also shows the participant proportion based on three areas is different (unbalanced). The various distribution proportions on the three areas potentially be a modifying factor. In table 2, we can see a significantly different dietary pattern between the three areas based on the X2 test. How the author control this issue?

Reply: We are thankful for you to point out the potential problems. Firstly, according to the statistics bureau, the distribution of birth population in different regions of Shaanxi province is uneven, and the sample size can basically represent the situation of the whole province. Secondly, we supplemented the results of regional stratification analysis to control for the influence of different areas on dietary patterns, as detailed in the supplementary tables. Finally, this study controlled for the impact of geographical differences through the geographically weighted regression (GWR) model, which was the focus of our study.

A.8. Table 2. Kindly show the number of samples n (%) to shows balanced samples in each category.

Reply: Thanks very much for your advice. We have made corresponding changes in Table 2.

A.9. Line 174 shows the different distribution of study participants. Has the author previously checked the various respondents' distribution in this area, probably influence this significance? Potential to be modified factor due to unbalanced distribution of samples, or consider reporting the effect in each region (stratification) in multivariate if the study investigates the association between risk factor of birth outcome.

Reply: Thank you very much for your question and valuable advice. We supplemented the results of regional stratification analysis to control for the impact of uneven distribution of dietary patterns on the risk of birth outcomes in different regions, as detailed in supplementary tables 1-3.

A.10. Table 2 and 3. For association purposes, the author can include adjusted p value from the multivariate analysis in this table with different mark, e.g ††.

Reply: Thanks very much for your advice. For the characteristics of dietary patterns among pregnant women in Shaanxi province, we included adjusted p-value from the multivariate analysis in supplementary table 4. For the baseline characteristics of the study population in Shaanxi province, univariate and multivariate analyses have been carried out in this study, as shown in Table 4, so repeat analysis is not considered.

A.11. Line 220. We suggest not to use “varied in line 220” because the proportion shows a slight difference.

Reply: Thanks for pointing out my problem. We have made corresponding modifications in the results section on Line 229.

A.12. Table 3. Refers to comment A10, to add p value multivariat analysis with different mark ††.

Reply: Thanks for your comments. The same circumstance with A10, the multivariate analysis as detailed in the supplementary Table 4.

A.13. Table 4. There is no socio-demographic characteristic was shown in this table, is only the dietary pattern that was a significant association with the birth outcome? kindly add in the complete information in the previous table for socio demographics characteristic for multivariate test.

Reply: We think your comment is very important for us to improve the manuscript. We mainly studied the relationship between dietary patterns and birth outcomes, socio-demographic characteristics were not the focus of this study. Considering the completeness of the results, the complete information of multivariate analysis was shown in Supplementary Table 5.

A.14. General comment for table 5. How is it far different using nonspatial logistic regression and GWR?

Reply: Thanks for your comments. We think your suggestion is very helpful. Due to the limitation of the data in this study, we only discussed the differences in the specific results of different regression models without further discussion. This was also a slight deficiency of this study, and we look forward to better improvement in the future.

A.15. Line 307 . Does it mean the different patterns mostly influenced by altitude? Why didn’t the author show others based on the rural and urban type of area.

Reply: Thanks for pointing out the question. The area altitude is only one part of the geographical environment factors that may influence the distribution of dietary patterns. The rural and urban types of the area have been eliminated in the previous data analysis process.

A.16. Line 354. Birth weight outcomes have close relation with the BMI of the mother (nutrition status of pregnant). Did the author consider this variable in their analysis or take in the limitation study.

Reply: Thanks very much for your question. Actually, this limitation should be considered while interpreting our results, although the analysis was adjusted for some socio-demographic confounders during pregnancy, the study was still subject to other potential confounders such as maternal nutrition status of pregnant (BMI). Considering the current situation of this study data, we have included this variable in the limitation study.

---

## [Decision Letter · Decision Letter 1]

7 Jul 2021

Geographical variations in maternal dietary patterns during pregnancy associated with birth weight in Shaanxi province, Northwestern China

PONE-D-20-40026R1

Dear Dr. Pei,

We’re pleased to inform you that your manuscript has been judged scientifically suitable for publication and will be formally accepted for publication once it meets all outstanding technical requirements.

Kind regards,

Frank T. Spradley

Academic Editor

PLOS ONE

Reviewers' comments:

Reviewer's Responses to Questions

**Comments to the Author**

1. If the authors have adequately addressed your comments raised in a previous round of review and you feel that this manuscript is now acceptable for publication, you may indicate that here to bypass the “Comments to the Author” section, enter your conflict of interest statement in the “Confidential to Editor” section, and submit your "Accept" recommendation.

Reviewer #1: All comments have been addressed

2. Is the manuscript technically sound, and do the data support the conclusions?

Reviewer #1: Yes

3. Has the statistical analysis been performed appropriately and rigorously? 

Reviewer #1: Yes

4. Have the authors made all data underlying the findings in their manuscript fully available?

Reviewer #1: Yes

5. Is the manuscript presented in an intelligible fashion and written in standard English?

Reviewer #1: Yes

6. Review Comments to the Author

Reviewer #1: Author have answered all of reviewer comments .

review

A.1 : ok, author has been changed in method section

A.2 : ok, enough explanation. The author has been controlled a potential reminder bias and they also add in the limitation study

A.3 : ok, PPS method sampling was used by the author and they have considered some criteria to show the representative area

A.4 : ok

A.5 : ok, the author have shown in the results section to give more clearly about geographical area description. line 180-183

A.6 : ok

A.7 : ok

A.8 : ok

A.9 :ok

A.10 : ok

A.11 : ok, the authors have made corresponding modifications

A.12 : ok

A.13 : ok

A.14 : ok

A.15 : ok

A.16 : ok, the author have explained in the end of discussion as limitation of study

7. PLOS authors have the option to publish the peer review history of their article (what does this mean?). If published, this will include your full peer review and any attached files.

Reviewer #1: No

---

## [Editor Report · Acceptance letter]

14 Jul 2021

PONE-D-20-40026R1 

Geographical variations in maternal dietary patterns during pregnancy associated with birth weight in Shaanxi province, Northwestern China 

Dear Dr. Pei:

I'm pleased to inform you that your manuscript has been deemed suitable for publication in PLOS ONE. Congratulations! Your manuscript is now with our production department. 

Kind regards, 

on behalf of

Dr. Frank T. Spradley 

Academic Editor

PLOS ONE